# Cellular Responses Required for Oxidative Stress Tolerance of the Necrotrophic Fungus *Alternaria alternata*, Causal Agent of Pear Black Spot

**DOI:** 10.3390/microorganisms10030621

**Published:** 2022-03-15

**Authors:** Miao Zhang, Yandong Zhang, Yongcai Li, Yang Bi, Renyan Mao, Yangyang Yang, Qianqian Jiang, Dov Prusky

**Affiliations:** 1College of Food Science and Engineering, Gansu Agricultural University, Lanzhou 730070, China; zhangmiao321@hotmail.com (M.Z.); zydlssy0701@163.com (Y.Z.); biyang@gsau.edu.cn (Y.B.); maorenyan1@163.com (R.M.); yanggasu2021@163.com (Y.Y.); jqianqian2022@163.com (Q.J.); 2Department of Postharvest Science of Fresh Produce, Agricultural Research Organization, Rishon LeZion 50250, Israel; dovprusk@volcani.agri.gov.il

**Keywords:** *Alternaria alternata*, oxidative stress, AsA-GSH cycle, thioredoxin system, redox balance

## Abstract

To establish successful infections in host plants, pathogenic fungi must sense and respond to an array of stresses, such as oxidative stress. In this study, we systematically analyzed the effects of 30 mM H_2_O_2_ treatment on reactive oxygen species (ROS) metabolism in *Alternaria alternata*. Results showed that 30 mM H_2_O_2_ treatment lead to increased O^2−^ generation rate and H_2_O_2_ content, and simultaneously, increased the activities and transcript levels of NADPH oxidase (NOX). The activities and gene expression levels of enzymes related with ascorbic acid-glutathione cycle (AsA-GSH cycle) and thioredoxin systems, including superoxide dismutase (SOD), catalase (CAT), glutathione reductase (GR), ascorbate peroxidase (AXP) and thioredoxin (TrxR), were remarkably enhanced by 30 mM H_2_O_2_ stress treatment. Additionally, 30 mM H_2_O_2_ treatment decreased the glutathione (GSH) content, whereas it increased the amount of oxidized glutathione (GSSG), dehydroascorbate (DHA) and ascorbic acid (AsA). These results revealed that cellular responses are required for oxidative stress tolerance of the necrotrophic fungus *A. alternata*.

## 1. Introduction

Zaosu pear (*Pyrus bretchneideri* cv. Zaosu) is greatly popular with consumers due to its thin skin, rich juicy flesh, abundant aroma, crispy texture, and high nutritional value [1]. However, pear fruits are peculiarly prone to postharvest diseases due to fungal infection [2]. Black pot caused by *Alternaria alternata* is the most serious postharvest disease of pear fruit [3]. *A. alternata* is a common genus of necrotrophic fungi that attack more than 100 plant species [4]. The pathogenic mechanisms of *A. alternata* in plants are varied and complex [5]. The host-specific toxins AK-Ⅰ and AK-Ⅱ are released by *A. alternata* during spore germination, which could restrain host defense systems or kill host cells [6,7]. The pathogen also secretes cell-wall-degrading enzymes during infection, which makes a great contribution to the destruction of the cell wall, producing suppressor and non-host-specific toxin metabolites, thereby accelerating the formation of black spot symptoms [8,9]. Recently, a dynamic balance of reactive oxygen species (ROS) was found playing an important role in fungal pathogenicity [4].

ROS, including superoxide (O_2_^−^), hydrogen peroxide (H_2_O_2_), hydroxyl radical (OH^−^), and singlet oxygen (^1^O_2_), are highly reactive, reduced forms of oxygen [10]. In general, a low concentration of exogenous or pathogen-derived ROS acts as a signal molecule to participate in programmed cell death and various stress responses, as well as to regulate the growth, development, and pathogenicity of the microorganism [11,12]. On the other hand, a high concentration of ROS causes oxidative cell damage; therefore, pathogens must strike a balance between these extremes by mitigating oxidative stress. Studies have reported that the growth and development of *Cryptococcus neoformans*, *Saccharomyces cerevisiae*, *S. pombe*, *Penicillium digitatum* of citrus, and *P. expansum* of apples were slowed to adapt to exogenous H_2_O_2_ stress [13,14,15]. 

During pathogen infection, plants rapidly generate ROS, resulting in an oxidative burst that limits pathogen spread through hypersensitive response (HR) or cell death [11,16]. In order to establish successful infections in host plants, pathogenic fungus have thus evolved many strategies to neutralize, scavenge, or repair the damage caused by ROS [17,18]. Evidence suggests that fungi, e.g., *Colletotrichum gloeosporioides*, *Magnaporthe grisea*, *Fusarium oxysporum*, and *Metarhizium acridum,* can scavenge excess ROS by using non-enzymatic systems, including the ascorbic acid-glutathione cycle (AsA-GSH cycle) and thioredoxin system, and enzymatic systems such as superoxide dismutase (SOD), catalase (CAT), and peroxidase (POD) [19,20,21]. H_2_O_2_, as an excellent intracellular redox signaling molecule, diffuses easily, and is relatively stable, which allows it time to interact with more targets compared with other ROS [22]. H_2_O_2_ treatment was reported to be able to activate the antioxidant systems of pathogenic fungi in vitro. In *M. oryzae*, H_2_O_2_ stress treatment upregulates the expression level of the antioxidant related genes *MoAPX3*, *NOX3*, *CATA*, *MoPRX1*, *TPX1* and *HYR1* [23]. In *Beauveria bassiana*, the activities of SOD and CAT enzymes significantly increase in the hyphae after H_2_O_2_ stress treatment [24]. Some studies have revealed the mechanisms of response to the oxidative burst in many fungi during the interaction between host and pathogen. Furthermore, previous studies suggest that there was strong tolerance to high concentration H_2_O_2_ in *A. alternata*. However, information is limited on the specific mechanism(s) deployed by *A. alternata* to maintain the intracellular redox balance under conditions causing oxidative stress. 

The aim of this study was to evaluate intracellular reactive oxygen generation, enzymatic activities related to the antioxidant system (AsA-GSH cycle and thioredoxin system), and gene expression of *A. alternata* in response to exogenous H_2_O_2_ stress treatment in vitro. We reveal how *A. alternata* activates the intracellular ROS scavenging system to maintain the intracellular redox balance after responding to oxidation burst. 

## 2. Materials and Methods

### 2.1. Fungal Strains and Culture Conditions

The *A. alternata* was originally isolated from a naturally decayed Zaosu pear fruit and characterized (KY397985.1) and was incubated on potato dextrose agar (PDA) at 28 °C. A spore suspension of *A. alternata* was configured according to Tang et al. [25], and the spore’s concentrations were determined by a hemocytometer. 

### 2.2. Sample Collection of the ROS Metabolism Evaluation

The spore suspension (2 µL, 1 × 10^6^ spores mL^−1^) of *A. alternata* was inoculated on PDA plates, cultured for 72 h, and then a 5 mm plug of mycelial agar was transferred to Cha’s liquid medium (2 g/L sodium nitrate, 1 g/L dipotassium hydrogen phosphate, 0.5 g/L magnesium sulfate, 0.5 g/L potassium chloride, 0.01 g/L ferrous sulfate, 30 g/L sucrose) containing 30 mM H_2_O_2_ for 0, 1, 2, 3, and 4 h. The hyphae of *A. alternata* was washed with sterile water and filtered with a Buchner funnel (Shanghai Laboratory Reagent Co., Ltd., Shanghai, China), and 1 g of the filtered mycelium was immediately used or frozen in liquid nitrogen and stored at −80 °C for the following assays [26]. 

### 2.3. Assessment of the O^2−^ Generation Rate and H_2_O_2_ Content of A. alternata

O^2−^ generation rate and H_2_O_2_ content were quantified as previously described by Zhang et al. [26], with a kit from Suzhou Comin Biotechnology (www.cominbio.com, accessed on 11 March 2020). The measure of O^2−^ generation rate was based on O^2−^ reactions with hydroxylamine hydrochloride to generate NO_2_^−^. NO_2_^−^ generates red azo compounds, which have a characteristic absorption peak at 530 nm under the action of p-aminobenzenesulfonamide and naphthalene ethylenediamine hydrochloric acid (Art.No: SA-1-G). The O^2−^ content in the sample was calculated according to the A530 value. The measure of H_2_O_2_ was based on H_2_O_2_ and titanium sulfate forming a yellow, titanium peroxide complex with characteristic absorption peak at 415 nm (Art.No: H_2_O_2_-1-Y). According to the manufacturer’s instructions, the 0.1 g of hyphae were collected to determine O^2−^ generation rate and H_2_O_2_ content using a spectrophotometer (UV-2450, Shimizu Company, Kyoto, Japan), against a standard curve; they were expressed as nmol/g·min and μmol/g FW, respectively. Assays were run in triplicates. 

### 2.4. ROS Metabolism Key Enzyme Activity and Antioxidant Substances Content of A. alternata Assay

NOX activity was based on the method previously described by Ge et al. [27], with minor modifications. A 1 g frozen hyphae from both the H_2_O_2_-treated and control group were suspended in 3 mL extract solution (25 mmol/L Mes-Tris, 250 mM sucrose, 3 mM ethylenedia-minetetraacetic acid (EDTA), 0.9% polyvinyl pyrrolidone (PVP), 5 mM dithiothreitol (DTT), and 1 mM phenylmethanesulfonyl fluoride (PMSF)) for 30 min in an ice bath, then the homogenates were centrifuged at 12,000× *g* for 30 min. The precipitate obtained was suspended in 1 mL supernatant and used for assaying NOX activity. The reaction mixture contained 50 mM Tris-HCl buffer (pH 7.5), 0.5 mM XTT, 100 µM NADPH, 30 µL enzyme extraction, with a final addition of NADPH. NOX activity was expressed as U/g FW (fresh weight), where 1 U = 0.01 A 600 min^−1^. 

The assays of SOD and CAT activities were based on the methods previously described by Ren et al. [28], with minor modifications. A 1 g frozen hyphae from both the H_2_O_2_-treated and control group were suspended in 3 mL 50 mM phosphate buffer (pH 7.5 containing 20 g/L PVP and 5 mM DTT), and fully ground in an ice bath. The supernatant was taken for testing after centrifugation (4 °C, 12,000× *g*, 30 min). SOD activity was expressed as U/g FW, where U was defined as the 50% inhibition of nitrotetrazolium chloride blue (NBT) by the enzyme at 560 nm. CAT activity was measured at 240 nm and expressed as nmol/min/g FW. 

The assay of GR and APX activities was based on the methods previously described by Sun et al. [29], with minor modifications. A 1 g frozen hyphae from both the H_2_O_2_-treated and control group were suspended in 3 mL 100 mM phosphate buffer (pH 7.5 containing 1 mM EDTA), and fully ground in an ice bath. The supernatant was taken for testing after centrifugation (4 °C, 12,000× *g*, 30 min). GR and APX activities were measured at 340 nm and 290 nm, respectively, and expressed as μmol/min/g FW. 

TrxR activity was quantified with a kit (Art.No: TRXR-1-W) from Suzhou Comin Biotechnology (www.cominbio.com, accessed on 11 March 2020). The TNB and NADP^+^ were generated by TrxR catalytic reduction in NADPH and DTNB. TNB has a characteristic absorption peak at 412 nm. TrxR activity was calculated according to the increase rates of TNB at the wavelength of 412 nm and expressed as nmol/min/g FW. The entire experiment was conducted three times.

The amounts of ASA and DHA were assayed by following the methods of Turcsanyi et al. [30]. In total, 0.5 g of frozen hyphae from both the H_2_O_2_-treated and control group was suspended in 2 mL 100 mM HCl, and fully ground in an ice bath. The supernatant was taken for testing after centrifugation (4 °C, 7800× *g*, 10 min). The ASA and DHA contents were measured at 216 nm and expressed as μmol/g FW.

The amounts of GSH and GSSG were assayed with a kit (Art.No: GSH-1-W/GSSG-1-W) from Suzhou Comin Biotechnology (www.cominbio.com, accessed on 11 March 2020). The GSH and GSSG contents were measured at 412 nm expressed as μmol/g FW. The thioredoxin concentration of *A. alternata* was measured by the thioredoxin concentration kit (Meilian Biotechnology Co., Ltd., Shanghai, China) and expressed as ng/L.

### 2.5. Real-Time qPCR Analysis

Total RNA was extracted from hyphae at different points after H_2_O_2_ treatment using the TRIzol reagent (QIAGEN, Shanghai, China) according to the manufacturers’ protocol. Reverse transcription was performed using 2 µg of RNA. GAPDH was used as an internal control. For qRT-PCR analysis, amplifications were performed using a Bio-Rad CFX96 real-time thermal cycler (Takara Biological Technology Co., Ltd., Dalian, China) and the QIAGEN QuantiNova TM SYBR^®^R Green PCR Kit (Takara Biological Technology Co., Ltd., Dalian, China). Three replicates were performed for each sample, and the relative transcript levels of *NOX, SOD, CAT, GR, APX* and *TrxR* were calculated using the 2^−^^ΔΔct^ method as described by Livak and Schmittgen [31]. The primers are shown in Table 1.

### 2.6. Statistical Analysis

Experimental data were expressed as the means ± the standard errors. Statistical analyses were performed with SPSS 19.0 (SPSS Inc., Chicago, IL, USA), and the data were tested by a one-way analysis of variance (ANOVA).

## 3. Results

### 3.1. O^2−^ Generation Rate and H_2_O_2_ Content of A. alternata Increased under Exogenous H_2_O_2_ Stress

In general, the O^2−^ generation rate of *A. alternata* treated by the exogenous H_2_O_2_ stress and the control group increased at first; then they steadily declined during 4 h of incubation, whereas O^2−^ generation rate in the H_2_O_2_-treated *A. alternata* significantly increased compared with the control group, peaking at 2 h after treatment, which was 22.9% higher than the control group (Figure 1A). 

As shown in Figure 1B, the H_2_O_2_ contents of the control group maintained at a relatively constant level during 4 h of incubation, whereas the H_2_O_2_ contents of *A. alternata* treated by the exogenous H_2_O_2_ stress increased at first and then declined, which were significantly higher than the control group. Peak H_2_O_2_ content was observed 2 h after treatment in H_2_O_2_-treated *A. alternata*, where its value was 41.4% higher than that of the control samples (*p* < 0.05).

### 3.2. NOX Activity and Gene Expression Level of A. alternata Increased after Exogenous H_2_O_2_ Stress

In comparison with the control group, nicotinimide adenine dinucleotide phosphate oxidase (NOX) activity increased constantly during the 2 h of incubation in 30 mM H_2_O_2_-treated *A. alternata*. Subsequently, it significantly declined in both control group and H_2_O_2_-treated *A. alternata*. The NOX activity of *A. alternata* treated by the exogenous H_2_O_2_ stress was always higher than that of the control group. Peak, at 2 h, was observed in the 30 mM H_2_O_2_-treated *A. alternata*, which was increased by 18.9% compared with the control group (Figure 2A). Further studies showed that H_2_O_2_ treatment significantly upregulated the gene expression levels of *NOX* at 2 h and 4 h of incubation (Figure 2B). Correspondingly, the *NOX* expression level of 30 mM H_2_O_2_-treated *A. alternata* was significantly increased by 97% compared with the control group after 2 h of incubation (*p* < 0.05).

### 3.3. SOD and CAT Activities, and Transcript Level of A. alternata Increased after Exogenous H_2_O_2_ Stress 

H_2_O_2_ treatments obviously increased the SOD activity of *A. alternata* from 1 to 4 h compared to the control group, with SOD activity peaking on 2 h and then slowly decreasing during later incubation of the H_2_O_2_-treated *A. alternata* (Figure 3A). In comparison with the control group, SOD activity of H_2_O_2_-treated *A. alternata* was increased by 33% after 2 h of incubation. Furthermore, *SOD* relative gene expression level of the *A. alternata* treated 30 mM H_2_O_2_ at 4 h upregulated by 10.8 times compared with the corresponding control group (Figure 3B).

As shown in Figure 3C, CAT activity of 30 mM H_2_O_2_-treated *A. alternata* raised steadily after 1 h of incubation, whereas SOD activity of the control group reduced rapidly from 2 to 4 h, and CAT activity of the H_2_O_2_ treatment group was significantly higher than that of the control group after 2 h of incubation. The highest level of CAT activity was observed at 4 h after the H_2_O_2_ treatments and was 1.8 times higher than that of the control group. Gene expression results showed *CAT* relative expression levels in H_2_O_2_-treated *A. alternata* were significantly higher than those of the control and peaked at 2 h (Figure 3D).

### 3.4. AsA-GSH Cycle System Involved in Maintaining the Intracellular Redox Balance of A. alternata after Exogenous H_2_O_2_ Stress

In general, the GR activity of *A. alternata* treated by the exogenous H_2_O_2_ stress rapidly increased at first and then gradually declined during 4 h of incubation, and the GR activity of the treatment group was significantly higher than the control group throughout the 4 h of incubation, with an increase of 61.6% at 1 h of incubation in contrast with the control (Figure 4A). Similarly, the maximum expression level of *GR* in H_2_O_2_-stress-treated *A. alternata* was observed at 1 h (Figure 4B) and shows an enhancement of 41.8 times in the H_2_O_2_-treated group compared with the control group (*p* < 0.05).

As shown in Figure 4, there was no significant difference in APX activity between the H_2_O_2_-treated group and the control during the entire incubation time, expect at 1 h (Figure 4C). The peak of the APX activity was discovered in the H_2_O_2_-stress-treated hypha at 1 h, which was raised by 69.5% after H_2_O_2_ treatment (*p* < 0.05). By contrast, H_2_O_2_ treatment obviously upregulated APX relative expression level during 4 h of incubation, which was 56.9 and 78.6 times higher than the control after 1 h and 2 h of incubation, respectively (Figure 4D). 

To further analyze the effect of H_2_O_2_ stress on the antioxidant system of *A. alternata*, the GSH and GSSG contents were determined during 4 h of incubation. The GSH content of 30 mM H_2_O_2_-stress-treated *A. alternata* increased continuously throughout the 4 h of cultivation, and the treatment group was significantly higher than the control group in the same period (Figure 4E). The high GSH content in the H_2_O_2_-treated *A. alternata* was observed at 4 h, which was accelerated by 26% compared to the control. Additionally, H_2_O_2_ treatment also caused a decline in GSSG content, which was reduced by 26.2% compared with the control after 4 h of cultivation (Figure 4F). 

As shown in Figure 4G, the ASA content of *A.*
*alternata* evidently increased after the 30 mM H_2_O_2_ treatment, and the highest content was detected at 1 h, which was increased by 30.4% in contrast with the control (*p* < 0.05). The DHA content in H_2_O_2_-treated *A. alternata* increased rapidly during 4 h of cultivation, and the treatment group maintained noticeably higher values than the control group over the entire cultivation period. Compared with the control, the DHA content was enhanced by 36.7% after 4 h of incubation (Figure 4H).

### 3.5. Thioredoxin System Involved in Maintaining the Intracellular Redox Balance of A. alternata after Exogenous H_2_O_2_ Stress

The thioredoxin concentration of 30 mM H_2_O_2_-treated *A. alternata* and the control group declined steadily, showing similar trends. However, H_2_O_2_ treatment significantly decelerated the decrease rate of thioresdoxin concentration of *A.*
*alternata*, which was 4.5% higher than the control group after 3 h of cultivation (Figure 5A).

The TrxR activity peaked on 2 h and then rapidly decreased during later incubation in 30 mM H_2_O_2_-treated *A. alternata*, and H_2_O_2_ stress effectively enhanced the TrxR activity of *A.*
*alternata* compared to the control group. After 2 h, the TrxR activity of treatment group was noticeably increased by 3.3 folds (Figure 5B). Similarly, the *TrxR* expression level of the H_2_O_2_-treated *A.*
*alternata* were markedly upregulated at 1 h and 2 h compared with the control group, which were increased by 53.3 folds and 56 folds, respectively (Figure 5C).

## 4. Discussion

ROS are the core of host plant defense, and also play a vital role in the successful invasion of host plants by pathogenic fungi [32]. During the disease resistance process of the host, the pathogen accumulates high levels of endogenous ROS, and this may create an imbalance in the cell redox state [33]. The data presented in this study showed that 30 mM H_2_O_2_ treatment significantly increased the O^2−^ generation rate and H_2_O_2_ content of *A. alternata* (Figure 1), indicating that oxidative stress induced the intracellular ROS production of *A. alternata*. Previous reports have demonstrated that NOX is the key enzyme source for the generation of ROS in pathogens, which use *FADH2* and two heme molecules as cofactors. In *P. expansum* and *Epichloë festucae*, the deletion of *NoxA* reduced the contents of O^2−^ and H_2_O_2_ [26,34]. In *A. alternata* of citrus, the deletion of *AaNoxA* increased sensibility to exogenous oxidants such as H_2_O_2_ and menadione [35]. In this study, we found that NOX activity and gene expression levels of *A. alternata* were significantly enhanced after it was 30 mM H_2_O_2_-treated (Figure 2). These results indicate that exogenous H_2_O_2_ treatment leads to initial accumulation of intracellular ROS via increasing NOX activity in *A. alternata*. However, pathogenic fungi can adapt to high concentrations of H_2_O_2_ and have a certain degree of tolerance to exogenous oxidative stress [23,36]. In our results, we found that *A.*
*alternata* can continue to grow after 30 mM H_2_O_2_ treatment (data not shown), indicating that there was strong tolerance to high concentrations of H_2_O_2_ in *A. alternata*.

In order to establish successful infections in host plants, pathogenic fungus can neutralize, scavenge, or repair the damage caused by ROS though activating its own antioxidant defense system [17,18]. The excessive ROS can be scavenged by sophisticated enzymatic systems such as SOD and CAT by pathogens [36]. In the present study, 30 mM H_2_O_2_ treatment significantly increased SOD activity, as well as the *SOD* gene expression levels, in *A. alternata* (Figure 3A,B), which may be conducive to the decrease in the content of O^2−^ and generation of more H_2_O_2_, since excess H_2_O_2_ is decomposed into H_2_O and O_2_ under the catalysis of CAT [37]. The observation in our study revealed that CAT activity and gene expression levels were also rapidly increased (Figure 3C,D), which was in accordance with the phenomenon in *Aspergillus niger*, where H_2_O_2_ treatment enhanced CAT activity [38]. In addition, the rapid increase in CAT activity proved it has a central role in counteracting the burst of ROS generated by the host cells to kill the invading fungal pathogens. 

GR is a flavoprotein oxidoreductase that may convert GSSG into GSH under the action of NADPH in the AsA-GSH cycle, whereas the GSH/GSSG ratio presents the redox state mediated by the intracellular glutathione system [39]. In this study, the GR activity and transcriptional levels were elevated by H_2_O_2_ treatment in *A.*
*alternata* (Figure 4A,B). Simultaneously, 30 mM H_2_O_2_ stress treatment increased GSH content but decreased GSSG content (Figure 4E,F). Similar results were also reported in *A. niger*, that H_2_O_2_ treatment increased the GSH/GSSG ratio [38]. However, in *P. chrysogenum*, there were no significant differences in GSH levels in response to H_2_O_2_ stress [40], indicating that different pathogens vary in their tolerance to H_2_O_2_ stress. APX is another key antioxidant enzyme in the AsA-GSH cycle, which breaks down H_2_O_2_ to form H_2_O and monodehydroascorbate (MDHA) that can be further converted to DHA [41]. The present results showed that H_2_O_2_ treatment induced the activity and transcriptional levels of APX in *A. alternata* (Figure 4C,D). In addition, the AsA and DHA contents were also enhanced after 4 h of incubation (Figure 4G,H). Results further confirmed that the AsA-GSH cycle system is involved in the *A. alternata* response to oxidative stress. In the *Alternaria* pathosystem, AsA-GSH cycle system may be activated immediately during the early infection stage, and scavenge ROS produced by the host to facilitate further infection of *A. alternata*

Thioredoxin (TRX) is a type of small molecule protein that can provide electrons for nucleotide reductase and peroxidase, which are involved in ROS scavenging to reduce oxidative damage [42]. In *Candida albicans*, the deletion of thioredoxin trx1 leads to increased sensitivity to H_2_O_2_ stress [43]. In this study, the thioredoxin concentration and TrxR activity were higher in H_2_O_2_-treated *A. alternata*, compared to the control group (Figure 5), indicating that the thioredoxin system is involved in maintaining the intracellular redox balance of *A. alternata*. However, Wang et al. [44] reported that the redox state of *Trichoderma reesei* was maintained via the GSH system, whereas Trtrx only plays a synergistic or backup role in oxidative stress resistance. Interestingly, in *Botrytis cinerea*, the thioredoxin system, not the GSH pool, is a central player in the maintenance of the redox status [45]. Therefore, the molecular mechanism of TRX in the response to oxidative stress in *A. alternata* needs further elucidation. 

## 5. Conclusions

Collectively, the present study showed that the activity of key enzymes and gene expression levels of ROS production systems in *A. alternata* increased initially upon oxidative stress, and then *A. alternata* effectively eliminated exogenous H_2_O_2_ and kept intracellular redox balance through increasing the activity of key enzymes and gene expression levels, and the antioxidant content of the AsA-GSH cycle and thioredoxin scavenging systems. The results suggested that *A. alternata* can effectively adapt to oxidative stress, but further work needs to be addressed for clarifying the underlying molecular mechanisms. 

## Figures and Tables

**Figure 1 microorganisms-10-00621-f001:**
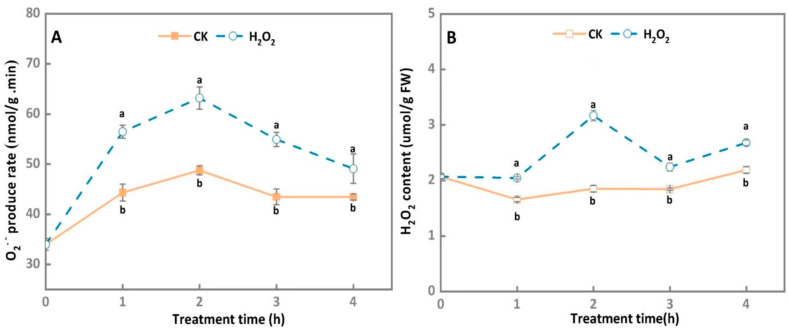
Effect of H_2_O_2_ stress treatment on O^2−^ generation rate (**A**) and H_2_O_2_ contents (**B**) of *A. alternata*. Different letters represent significant differences (*p* < 0.05), bars indicate standard error (±SE).

**Figure 2 microorganisms-10-00621-f002:**
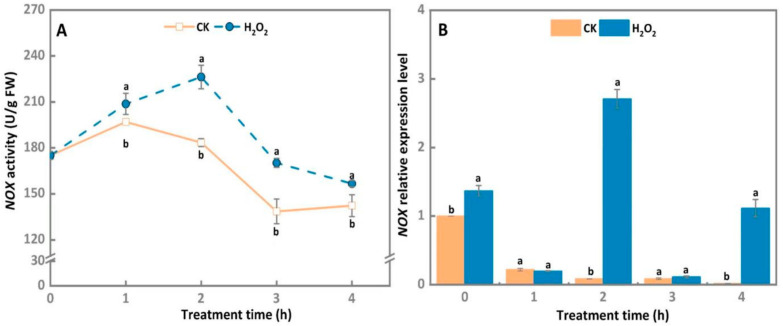
Effect of H_2_O_2_ stress treatment on NOX activity (**A**) and gene expression level (**B**) of *A. alternata*. GAPDH was used as an internal control; different letters represent significant differences (*p* < 0.05), bars indicate standard error (±SE).

**Figure 3 microorganisms-10-00621-f003:**
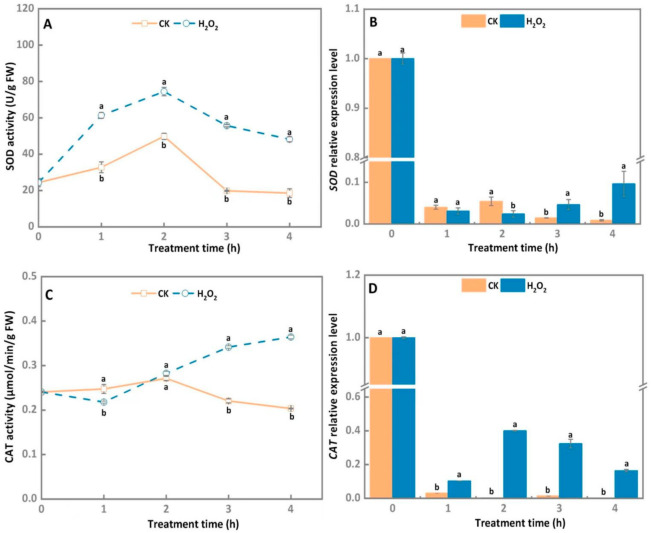
Effect of H_2_O_2_ stress treatment on SOD (**A**) and CAT activities (**C**), and gene expression level (**B**,**D**) of *A. alternata*. *GAPDH* was used as an internal control; different letters represent significant differences (*p* < 0.05), bars indicate standard error (±SE).

**Figure 4 microorganisms-10-00621-f004:**
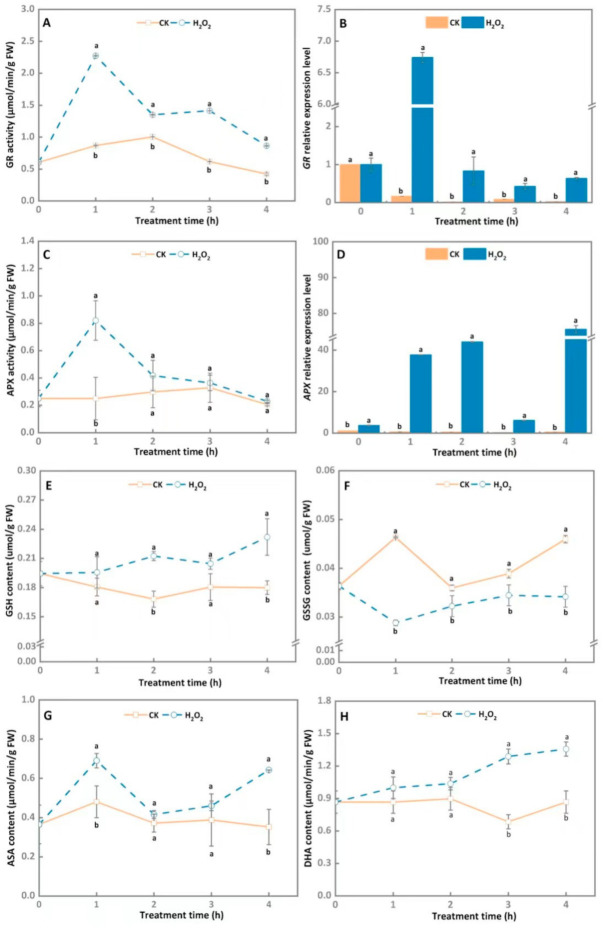
Effect of H_2_O_2_ stress treatment on AsA-GSH cycle system related enzymes activities and gene expression, as well as related metabolite contents of *A. alternata* (**A**: GR activity; **B**: GR relative expression; **C**: APX activity; **D**: APX relative expression; **E**: GSH content; **F**: GSSG content; **G**: ASA content; **H**: DHA content). GAPDH was used as an internal control; different letters represent significant differences (*p* < 0.05), bars indicate standard error (±SE).

**Figure 5 microorganisms-10-00621-f005:**
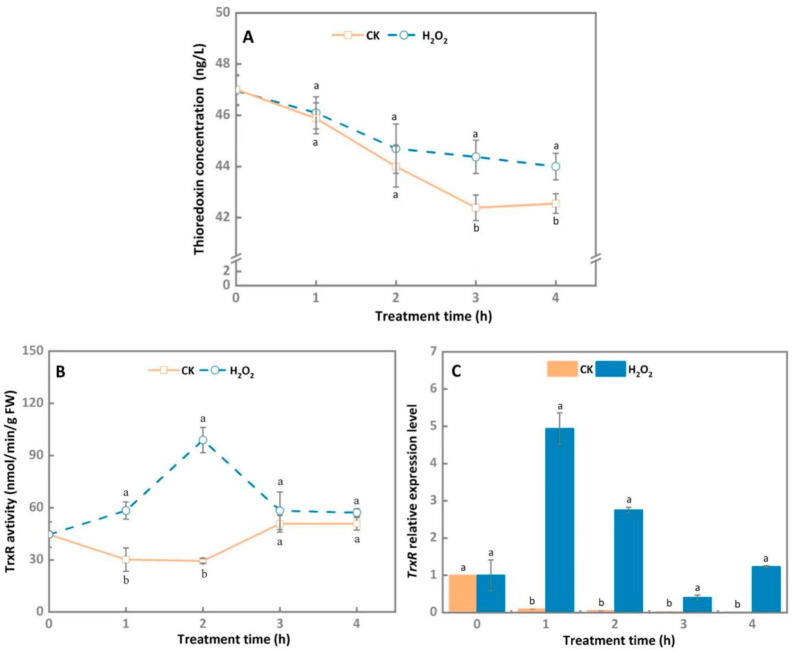
Effect of H_2_O_2_ stress treatment on thioredoxin concentration (**A**), TrxR activity (**B**) and gene expression level (**C**) of *A. alternata*. *GAPDH* was used as an internal control; different letters represent significant differences (*p* < 0.05), bars indicate standard error (±SE).

**Table 1 microorganisms-10-00621-t001:** Sequence of primer design.

Gene	Sequence of Primer (5′ to 3′)	TM
*NOX*	F: AACGAAGTCGCAGTTCTTATTGR: GGCGGAGGTGGTAGATAGATT	60
*SOD*	F: AACAACTTCAGCGAGCAAATCR: TTGATGGCAGCAGATAGCG	60
*CAT*	F: CCACGGCACCTTTGTTTCTR: ATCTCGCACTGTGTCAGCACT	60
*TrxR*	F: GCGGTATCGTCAGGCTATCAR: CTATCCCTATGCTGTCATCTTGC	60
*GR*	F: GCCAAACACGGTGCAAAAGTR: TTTGAAGGTCTCGGCAATCG	60
*APX*	F: AATGCTGGTCTCAAGGCTGCR: GACCCTGCATCTCCTGGATG	60

## Data Availability

The data presented in this study are available upon request from the corresponding authors.

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
