# Peer review of "Cellular Responses Required for Oxidative Stress Tolerance of the Necrotrophic Fungus Alternaria alternata, Causal Agent of Pear Black Spot"

_microorganisms, 2022, doi:10.3390/microorganisms10030621_

Round 1

Reviewer 1 Report

I have an opportunity to review manuscript :” Cellular responses required for oxidative stress tolerance of the necrotrophic fungus Alternaria alternata, causal agent of pear black spot” submitted to Microorganisms MDPI journal.

Authors concentrated on  effects of 30mM H2O2 treatment on reactive oxygen species (ROS) metabolism in necrotroph Alternaria alternata systemic infection. Analysed data indicated that 30mM H2O2 treatment leads to increase O2·- generation rate and H2O2 content, simultaneously, increase the activities and transcript levels of NADPH oxidase (NOX).

Furthermore, 30mM H2O2 treatment decreased the glutathione (GSH) content, while increased the amount of oxidized glutathione (GSSG), dehydroascorbate (DHA) and ascorbic acid (AsA).

I understand that as Authors stated: “However, information is limited on the specific mechanism(s) deployed by A. alternata to maintain the intracellular redox balance under conditions causing oxidative stress”. Despite of it, I should ask about novelty aspect in this manuscript??, because when Authors treat plant in biotic or abiotic stress with hydrogen peroxide the results are expected- it means it was expected tendency that all analysed enzymes will be induced; Therefore, Authors should convince the reader about novelty aspect of obtained results;

Furthermore, I ask about the Authors hypothesis?, because in my opinion the results will or are predictable;

I understand Authors statement that, “In addition, the AsA and DHA contents were also enhanced after 4 h of incubation (Fig. 4G; H), results further confirmed that the AsA-GSH cycle system is involved in A. alternata response to oxidative stress: but what kind of function can have glutathione metabolism tendency [like Authors presented] in Alternaria pathosystem?

In materials and methods is too many understatements, if enzymes activity and expression were analyzed direct from hypha or spores please underlined it in M&M section, if in some aspect pear tissues were used, please also indicate it; I guess that it was all samples analysed or collected from hyphae;

How to concluding between Alternaria symptoms induction and the pool of enzymes induction?

How to explain that SOD relative expression were induced and then in 1 and 2 h decreased?

Author Response

Thank you very much for giving us such good suggestions about revising this paper. According to your suggestions and comments, we have earnestly revised this paper. The details of the changes made during revision are as follow:

Point1: I understand that as Authors stated: “However, information is limited on the specific mechanism(s) deployed by A. alternata to maintain the intracellular redox balance under conditions causing oxidative stress”. Despite of it, I should ask about novelty aspect in this manuscript?, because when Authors treat plant in biotic or abiotic stress with hydrogen peroxide the results are expected- it means it was expected tendency that all analysed enzymes will be induced; Therefore, Authors should convince the reader about novelty aspect of obtained results;

Furthermore, I ask about the Authors hypothesis?, because in my opinion the results will or are predictable;

Response 1: Previous studies have reported that pathogenic fungi such as Botrytis cinerea, Magnaporthe oryzae, can effectively resist the higher levels of reactive oxygen species and maintain intracellular redox balance and pathogenicity through sensitive response mechanisms and their own ROS scavenging system (Viefhues et al., 2014; Samalova et al., 2014). Therefore, deoxidation and maintenance of intracellular redox balance is necessary for the successful infection and colonization of A. alternata. However, how does A. alternata respond to oxidative stress and effectively scavenge exogenous ROS? The main ROS scavenging system of A. alternata needs to be further elucidated.

In this study, exogenous H2O2 stress effectively activated the AsA-GSH cycle and thioredoxin systems and then maintained redox balance of A. alternata. This study confirmed that the specific mechanism of A. alternata in response to the higher level of reactive oxygen species produced by the host in vitro, our results presented in this study may be useful for widening and further dissecting the molecular mechanisms of A. alternata, the causal agent of black spot in pear fruit, in response to oxidative burst in host-pathogen interaction.

Point2: I understand Authors statement that, “In addition, the AsA and DHA contents were also enhanced after 4 h of incubation (Fig. 4G; H), results further confirmed that the AsA-GSH cycle system is involved in A. alternata response to oxidative stress: but what kind of function can have glutathione metabolism tendency [like Authors presented] in Alternaria pathosystem

Response 2: As the reviewer’s suggestion, we have added corresponding content. “In addition, the AsA and DHA contents were also enhanced after 4 h of incubation (Fig. 4G; H), results confirmed that the AsA-GSH cycle system is involved in A. alternata response to oxidative stress. In Alternaria pathosystem, AsA-GSH cycle system may be activated immediately during early infection stage, and scavenge ROS produced by the host to facilitate further infection of A. alternata”.  

Point3: In materials and methods is too many understatements, if enzymes activity and expression were analyzed direct from hypha or spores please underlined it in M&M section, if in some aspect pear tissues were used, please also indicate it; I guess that it was all samples analysed or collected from hyphae;

Response 3: As the reviewer’s suggestion, we totally agree and carefully revised enzyme activity assays and samples collected in the M&M section. In present study, all samples were collected from hyphae of A. alternata.

Point 4: How to concluding between Alternaria symptoms induction and the pool of enzymes induction?

Response 4: Our previous results indicated that A. alternata has the ability to relieve the damage of exogenous ROS. The results presented in this study showed that the related enzymes and antioxidants in AsA-GSH cycle and thioredoxin systems were induced by exogenous ROS stress. These data suggest these related enzymes may be involved in the early infection process of A. alternata through scavenging ROS produced by the host. But their specific mode of action need to be further explored in the future.

Point 5: How to explain that SOD relative expression were induced and then in 1 and 2 h decreased?

Response 5: Thank you for your careful review. SOD relative expression actually decreased in 1 and 2 h after our multiple repeated tests in Fig. 3B, our explanation that CAT (Fig. 3B) and NOX (Fig. 2B) genes expression were firstly induced by exogenous H2O2  stress for degrading H2O2 and producing O2-, and then SOD relative expression were induced to degrading O2-.

Reviewer 2 Report

The authors have designed a paper to study an aspect of great interest in the plant-pathogen relationship and its success in the infection process. The experiments are well designed and the number of results obtained are of great value to obtain conclusions of interest. I think it can be published in Microorganisns after making some modifications and clarifying some issues.

1. Why have the authors chosen 30 mM H2O2?

2. My main complaint about the work is the material and methods. Although normally it is not given the importance it deserves, this section is very important to me. All the necessary details must be given so that any researcher can repeat the experiment. I think that all of us abuse when using references from other authors to not detail the methods, and I think that this should be avoided. Obviously it is necessary to cite the author from which a certain method has been obtained, but it does not exempt from explaining it. For example:

23. Assessment of the O2·- generation rate and H2O2 content of A. alternata.

I have not been able to find the kit that the authors have used to assess these parameters. Please explain what the measure is based on and give a link where you can find the kit.

2.4. ROS metabolism key enzyme activity and antioxidant substances content of A. alternata 95 assay

In this section it is necessary to explain much better how enzyme assays are done, but what is worse is that the authors say that to measure GSH, GSSG, ASA and DHA they followed the method of Turcsanyi et al. When I have gone to see that work, it turns out that those authors did not measure GSH and GSSG. This can not be.

Regarding the results, given that the journal is an open access and admits figures in color, I think that the authors should make more elaborate figures, with color.

Author Response

Thank you very much for giving us such good suggestions about revising this paper. According to your suggestions and comments, we have earnestly revised this paper. The details of the changes made during revision are as follow:

Point1: Why have the authors chosen 30 mM H2O2?

Response1: Preliminary experimental results showed that A. alternata can scavenge exogenous H2O2 after 30mM H2O2 stress treatment. (as the figure shows)

Point2: My main complaint about the work is the material and methods. Although normally it is not given the importance it deserves, this section is very important to me. All the necessary details must be given so that any researcher can repeat the experiment. I think that all of us abuse when using references from other authors to not detail the methods, and I think that this should be avoided. Obviously it is necessary to cite the author from which a certain method has been obtained, but it does not exempt from explaining it. For example:

Assessment of the O2·- generation rate and H2O2 content of A. alternata. I have not been able to find the kit that the authors have used to assess these parameters. Please explain what the measure is based on and give a link where you can find the kit.

Response2: As the reviewer’s suggestion, we totally agree and carefully revised M&M section. O2·- generation rate and H2O2 content were quantified as previously described by Zhang et al. [26], with a kit from Suzhou Comin Biotechnology (www.cominbio.com). The measure of O2·- generation rate was base on O2·- reacts with hydroxylamine hydrochloride to generate NO2-, NO2- generates red azo compounds which has a characteristic absorption peak at 530nm under the action of p-aminobenzenesulfonamide and naphthalene ethylenediamine hydrochloric acid. The O2·- content in the sample was calculated according to the A530 value. The measure of H2O2 was base on H2O2 and titanium sulfate form yellow titanium peroxide complex with characteristic absorption peak at 415nm.

Point3: 2.4. ROS metabolism key enzyme activity and antioxidant substances content of A. alternata assay. In this section it is necessary to explain much better how enzyme assays are done, but what is worse is that the authors say that to measure GSH, GSSG, ASA and DHA they followed the method of Turcsanyi et al. When I have gone to see that work, it turns out that those authors did not measure GSH and GSSG. This can not be.

Response3: As the reviewer’s suggestion, we totally agree and carefully revised enzyme assays in the M&M section.

Point4: Regarding the results, given that the journal is an open access and admits figures in color, I think that the authors should make more elaborate figures, with color.

Response4: As the reviewer’s suggestion, the figures was modified to color.

Round 2

Reviewer 1 Report

In my opinion Authors improved significantly the manuscript;

Material and methods section were extensively, better described as well as Authors hypothesis seems to be clear in revised version of the manuscript. Almost all my suggestion Authors took into account, I would like to underlined that maybe I am not agree with all opinion, but the explanation are rather convinced-expecially aspect of results novelty;